# Structural and Functional Dissection of the 5′ Region of the *Notch* Gene in *Drosophila melanogaster*

**DOI:** 10.3390/genes10121037

**Published:** 2019-12-12

**Authors:** Elena I. Volkova, Natalya G. Andreyenkova, Oleg V. Andreyenkov, Darya S. Sidorenko, Igor F. Zhimulev, Sergey A. Demakov

**Affiliations:** 1Department of the Structure and Function of Chromosomes, Laboratory of Chromosome Engineering, Institute of Molecular and Cellular Biology SB RAS, 630090 Novosibirsk, Russia; volk@mcb.nsc.ru (E.I.V.); anata@mcb.nsc.ru (N.G.A.); andreenkov@mcb.nsc.ru (O.V.A.); demidova.daria@mcb.nsc.ru (D.S.S.); zhimulev@mcb.nsc.ru (I.F.Z.); 2Structural, Functional and Comparative Genomics Laboratory, Novosibirsk State University, 630090 Novosibirsk, Russia

**Keywords:** *Drosophila melanogaster*, *Notch* gene, CRISPR/Cas9 system, gene activity regulation, insulator proteins, polytene chromosomes, open chromatin state, interbands

## Abstract

Notch is a key factor of a signaling cascade which regulates cell differentiation in all multicellular organisms. Numerous investigations have been directed mainly at studying the mechanism of Notch protein action; however, very little is known about the regulation of activity of the gene itself. Here, we provide the results of targeted 5′-end editing of the *Drosophila Notch* gene in its native environment and genetic and cytological effects of these changes. Using the Clustered Regularly Interspaced Short Palindromic Repeats/CRISPR associated protein 9 (CRISPR/Cas9) system in combination with homologous recombination, we obtained a founder fly stock in which a 4-kb fragment, including the 5′ nontranscribed region, the first exon, and a part of the first intron of *Notch*, was replaced by an attachment Phage (attP) site. Then, fly lines carrying a set of six deletions within the 5′untranscribed region of the gene were obtained by *ΦC*31-mediated integration of transgenic constructs. Part of these deletions does not affect gene activity, but their combinations with transgenic construct in the first intron of the gene cause defects in the Notch target tissues. At the polytene chromosome level we defined a DNA segment (~250 bp) in the *Notch*5′-nontranscribed region which when deleted leads to disappearance of the 3C6/C7 interband and elimination of CTC-Factor (CTCF) and Chromator (CHRIZ) insulator proteins in this region.

## 1. Introduction

The *Notch* pathway is an important and evolutionarily conserved signaling system involved in the development of multicellular organisms [1,2,3]. Similarly to other signaling pathways, it determines the fate of differentiating cells throughout development [4,5]. Notch is a transmembrane receptor protein that in *Drosophila melanogaster* participates in the morphogenesis of both the central and peripheral nervous system, and the development of the eye, wing, and segmented appendages such as legs, antennae, and muscles. During morphogenesis of the central nervous system and sensory bristles in *Drosophila* embryos, the *Notch* signaling cascade serves to separate neural and epidermal primordia, transmitting presumptive nerve cell signals that prevent neighboring cells from differentiating into the nervous tissue [6,7]. The *Notch* signaling pathway regulates local interactions between cells during eye formation. Decreased Notch activity causes differentiating retinal cells to choose an unusual development pathway, thereby leading to the formation of defective photoreceptors, and altering the number and location of the eye and bristle constituent elements [8,9]. It was shown that the *Notch* signaling pathway is involved in the interactions between neighboring cells at the dorso-ventral border of the wing imaginal disc at the proliferation stage, and in the formation of the wing margin [10]. *Notch* signaling is required for the mitotic-to-endocycle transition in the follicle cells of the *Drosophila* egg chamber [11] and for the proliferation and differentiation of follicle cells during *Drosophila* oogenesis [12]. Local *Notch* expression is necessary for the formation of leg compartments and ligaments between their segments, and determining the boundaries of leg compartments [13].

In the *D. melanogaster* genome, the *Notch* (*N*) gene is located on the X chromosome. Fine cytogenetic analysis of chromosomal rearrangements and other mutations showed that on the polytene chromosome map this locus is situated in the 3C7 band [14,15]. Later, using high-resolution in situ hybridization, this band was found to encompass mainly the structural part of the gene, with the 5′-end of this gene located at the decompacted region of the chromosome, which corresponds to the interband 3C6/C7 [16]. The transcribed portion of the gene occupies about 37 kb, and produces transcripts about 10 kb long [17].

Hundreds of mutations have been obtained for the *Notch (N)* locus (http://www.flybase.org/reports). A significant part of these mutations affect the coding part of the gene, causing development disorders of organs of neuro-ectodermal origin—eyes, wings, bristles, and gonads. Mutations in the introns of the gene that affect the fly development have also been obtained and characterized. These include, for example, mutations of the *facet* group caused by insertions of *copia*-like mobile elements into the second intron of the gene [18]. All mutations in this group affect the correct formation of the facet eye. The fact that introns contribute to the regulation of *Notch* activity was also demonstrated in transgenic experiments with constructs containing either the full-length genomic copy of the gene or the minigene versions that lacked most of the introns [17].

Nonetheless, it should be noted that very few mutations have been characterized that would affect the 5′ regulatory region of the gene. One of these mutations, *N^fa-swb^*, a recessive mutation affecting eye development, is a deletion that localizes immediately upstream of the *Notch* promoter region and spans a DNA segment of 880 bp [17]. Negative influence of this mutation on *Notch* activity can be suppressed *in cis* combinations with certain chromosomal rearrangements involving more distal sequences of the gene [15,19,20]. It was suggested that the suppression effects occur due to *N^fa-swb^* that disrupts or inactivates a domain boundary. Later it was shown that the DNA fragment affected by this deletion exhibits insulator properties. Namely, it can protect reporter transgenes from position effects and blocks enhancer–promoter communication [21].In different tissues, the proximal part of this DNA region has various DNase I-hypersensitive sites [21,22], whose presence is characteristic of many insulator elements [23,24]. Using whole genome data on the distribution of chromatin proteins and other characteristics of chromatin in *D. melanogaster* cell lines [25,26,27], areas enriched with dCTCF, Centorsomal protein 190 kD (CP190), and GAGA Factor (GAF) insulator proteins, as well as with open chromatin marks such as Chromator (CHRIZ) protein, RNA polymerase II, and chromatin remodeling proteins (ISWI, NURF301, dRING) were identified in the 5′ region of the *Notch* gene. In this area, nucleosome density is reduced, and histone H1 is depleted [28]. Several independent studies on the genome-wide chromatin conformation capture (Hi-C) from embryos and *Drosophila* cell cultures have shown that the *Notch* gene body is located within one of the topological associated domains (TADs), and the 5′ regulatory region of the gene corresponds precisely to the TAD boundary [29,30,31]. Taken together, these data indicate that the chromatin of promoter region of the *Notch* gene is in the “open” state, even in those tissues where this gene is inactive and, apparently, contains sequences that perform barrier functions.

In the present work, using a combination of modern techniques of directed genome editing, we conducted a dissection of the 5′ region of the *Notch* gene to identify the sequences involved in the regulation of activity of this complex locus. First, we applied CRISPR/Cas9 system in combination with the homologous recombination method [32] to create a founder fly strain in which about 4 kb of DNA containing the promoter region, the first exon, and a part of the first intron of the *Notch* gene were replaced by an attachment Phage (attP) site. Further, using the *ΦC*31 phage attP/attB system, we placed the deleted DNA fragment back, which led to the “rescue” of lethality of flies containing the basic deletion. Next, we obtained a series of directed deletions affecting possible functional regions of the 5′ regulatory region of the gene and analyzed the phenotypes of these deletions. The scheme for obtaining deletions was such that at the intermediate stage, all lines after the *ΦC*31-induced integration additionally contained the insertion of auxiliary elements of the transgenic construct in the first intron. It turned out that this insertion significantly influenced the manifestation of phenotypes.Situated *in cis* with the resulting deletions it caused various defects in eye morphology, and in transheterozygote with the basic deletion it caused a pleiotropic effect. The strongest effects were observed when combining the insertion with deletions that remove the sequence of the putative insulator.

## 2. Materials and Methods

### 2.1. Fly Stocks

As a source of the Cas9 nuclease we used the fly stock *w1118; PBac{vas-Cas9,U6-tracrRNA}VK00027* (#51325 in Bloomington Drosophila Stock Center, BDSC). The *yw nanos*-Cre stock [33] contained a source of Cre-recombinase. We used *w^1118^sov^ML150^/FM0*, FM7a, and *P{Tb^1^}FM7-A* balancer stocks, the *w^a^N^fa-swb^*stock (Bloomington Drosophila Stock Center (BDSC)), and the Oregon-R (OR) wild type stock from the laboratory collection.

### 2.2. DNA Clones

The cos163A10 clone from the cosmid library (Lorist 6) of the *D. melanogaster* genome was kindly provided by Dr. I. Siden-Kiamos (Greece).

### 2.3. Donor Construction for Homologous Recombination (HR)

Using PCR on the cosmid cos163A10 DNA template containing a segment from the *Notch Drosophila* locus, homology arms 5′-HA (2381 bp) and 3′-HA (3510 bp) flanking the DNA fragment of 4035 bp (N-4k) from the 5′-end of the *Notch* gene were synthesized (Figure 1B). For this purpose, primer pairs with restriction sites *Not*I and *Kpn*I for 5′-HA (5HA(Not)f: gcggccgccattagtgatgatagtacatg, 5HA(KpnI)r: ggtaccatatggagcggtcgttgtcta) and *Asc*I and *Stu*I for 3′-HA (3HA(Asc)f: ggcgcgccatatggagcggtcgttgtc, 3HA(Stu)r: gcccaaaggcctggaagaca) were used (hereinafter, the orientation of primers is 5′-3′). The fragments were sequentially integrated into the pGX-attP(pRK-2) vector digested at these sites [34]. As a result, pGX-attP {5′-3′ HA-N} donor construct containing 5′ and 3′ homology arms, the mini-*white* reporter gene with GMR (gene modulating RNase II) enhancer surrounded by *loxP* sites for Cre-mediated recombination, and an attP site was obtained (Figure 1B).

### 2.4. CRISPR/Cas9 Mediated Homology Directed Reparation and Synthesis of a Founder Line

To conduct homologous recombination using the CRISPR/Cas9 system in the 5′ region of the *Notch* gene, sequences for target-specific guide-RNAs (gRNAs) were selected. The online resource fly CRISPR Optimal Target Finder tool (http://tools.flycrispr.molbio.wisc.edu/targetFinder/) was used for this purpose. Then, using the method «Generating targeting chiRNAs» (http://flycrispr.molbio.wisc.edu) two constructs containing the corresponding gRNAs were created based on the pU6-BbsIchiRNA vector (Plasmid #45946, Addgene). Oligonucleotide sequences for N-gRNA1 were cttcgagctcgatgagtttcgcag/aaacctgcgaaactcatcgagctc, and for N-gRNA2: cttcggagagttcgcttccaaagc/aaacgctttggaagcgaactctcc.

Embryos of the line with the endogenous source of Cas9 nuclease (y^1^M{vas-Cas9,U6-tracrRNA}ZH-2Aw^1118^) were microinjected with a mixture of the plasmids pU6-N-gRNA1 and pU6-N-gRNA2 (100 ng/µL each) and the donor plasmid pGX-attP{5′-3′ HA-N} (500 ng/µL). Microinjection into embryos was performed via standard procedures as described previously [35]. As a result of the transformation, 120 flies were obtained from 198 treated embryos. Among the 17,000 offspring, four red-eyed females were found, two of which were sterile. Two fertile females became the founders of stocks in which the reporter gene [w+] was linked to the X chromosome, and the red-eyed males did not survive. After removing the [w+] reporter flanked by the *loxP* sites using Cre recombinase, the founder stock with a genotype *y w dN[w*–*]/FM0* was obtained, in which about 4 kb from the 5′ region of the *Notch* gene (centered at the gene transcription initiation site) were replaced by the attP site (Figure 1C,D). The presence of deletions and excision accuracy of the loxP-flanked reporter [w+] were confirmed by PCR and sequencing (Appendix A).

### 2.5. Notch Transgenic Constructs

N-4k construct: To obtain an intact fragment (N-4k) encompassing the 5′ end of the *Notch* gene (Figure 1B), PCR was performed on the DNA template of the cosmid cos163A10 with the primers N4f (cgcagcggcaaattatatc) and N4r (cgaccgctccatatgcaaatac) followed by PCR with the modified primers N4k(Not)f (**gcggc**cgcagcggcaaattatatc) and N4k(Asc)r (**ggcgcg**ccgctccatatgcaaatac). Finally, the N-4k PCR fragment and pGE-attB-GMR vector DNA (Drosophila Genomics Resource Center (DGRC)) were digested with NotI and AscI and ligated together to form the pGE-attB-GMR[N-4k] construct (Figure 1D).

Deletions in the *Notch* promoter region. Deletions were produced using the PCR primer pairs indicated below and the pGE-attB-GMR[N-4k] construct as a template. PCR products were isolated using the QIAquickPCR Purification Kit (QIAGEN, Inc, Germantown, MD, USA), then ligated by Gibson Assembly MasterMix (NEB, E2611S), followed by PCR amplification using the primers N-df(ClaI) N-dr(XhoI). Hybrid DNA fragments harboring the deletions of interest were digested with ClaI and XhoI and cloned into the pGE-attB-GMR[N-4k] vector (from which the intact 1770-bp ClaI-XhoI DNA fragment was removed).

The *d1 construct* was obtained using the primers N-df(ClaI)–taatcgataatccccaagccg/N-d1r**- catagcgttttttcgcaa**ctccgcttcc and N-d1f**-ttgcgaaaaaacgctatg**acagcactaaagc/N-dr(XhoI)–cgactcgaggtagatacac.

The *d2 construct* was obtained using the primers N-df(ClaI)–taatcgataatccccaagccg/N-d2-r**–atagcgtttgattgaaaaa**cgcacttgtagca and N-d2-f**-ttttcaatcaaacgctat**gacagcactaaagc/N-dr(XhoI)–cgactcgaggtagatacac.

The *d3 construct* was obtained using the primers N-df(ClaI)–taatcgataatccccaagccg/N-d3-r**–gggagccatttgattgaaa**aacgcacttgtagca and N-3d-f**-tttcaatcaaatggctccc**cgccatac/N-dr(XhoI)–cgactcgaggtagatacac.

The *d4 construct*was obtained by removing the PacI-ClaI segment from the N-4k fragment in the pGE-attB-GMR[N-4k] vector.

The *dfa-swb construct.*Several DNA amplification reactions were conducted. First, PCR reaction was performed using the primers N-df(ClaI)/fa-dist-r (taatcgataatccccaagccg/gtagatcgctcaaatgttttac). Next, two-step PCR using fa-prox-f1/N-dr(XhoI) (cgatctacaatggctccccgccatac/cgactcgaggtagatacac) primers (step 1) and fa-prox-f2/N-dr (XhoI) primers (**catttgagc**gatctacaatggctc/cgactcgaggtagatacac) (step 2) was carried out. The fragments N-df(ClaI)/fa-dist-r and fa-prox-f2/N-dr(XhoI) were ligated by using the Gibson Assembly Master Mix (NEB, #E2611) followed by PCR, digestion with *Cla*I and *Xho*I restriction enzymes, and cloning into the vector pGE-attB-GMR[N-4k].

To obtain the *dfa-swb^LK^* on a template of genomic DNA of the *w^a^N^fa-swb^*fly stock, PCR was conducted using the N-df(ClaI) and N-dr(XhoI) primers. Next, the amplified fragment containing the classical fa-swb deletion was similarly cloned into the pGE-attB-GMR[N-4k] vector.

The localization of deletions on the physical map of the *Notch* gene 5′ region is shown in the Appendix A.

The *N-resc-dGMR* construct was obtained based on the vector pGE-attB-GMR[N-4k]. First, a DNA fragment containing the GMR-hsp70-mini-white segment was removed from this vector by digestion with *Bsi*WI and *Sac*I (Figure 1D). Then, using the primers dGMR(BsiWI)/w(SacI) (agcagtcgtCgtacGtacgtcatg/gtcaaagagctcaaacagctcgg), the DNA fragment was amplified that contained only the *hsp70* promoter and the 5′ segment of mini-white gene and had *Bsi*WI and *Sac*I restriction sites on the flanks. Using these sites, the fragment was cloned into the pGE-attB-GMR[N-4k] vector.

At all steps of construct assembly and after their integration into the genome, the functional elements were sequenced using BigDye Terminator Cycle Sequencing (AppliedBiosystems, Thermo Fisher Scientific, Waltham, MA, USA).

### 2.6. Immunostaining of Polytene Chromosomes

Salivary glands were dissected from third-instar larvae reared at 25 °C. Standard procedure for chromosome squashing was followed. Immunostaining was done as described previously [36], with minor modifications. Dilutions of the primary antibodies were as follows: rabbit polyclonal anti-CHROMATOR (kindly provided by Dr. A.A. Gorchakov) (1:600) and rabbit anti-dCTCF (kindly provided by Prof. P.G. Georgiev) (1:150). Chromosome squashes were then incubated with secondary FITC-labeled goat anti-rabbit IgG-specific conjugates (Abcam, 1:200). Chromosomes were examined using epifluorescence optics (Olympus BX50 microscope) and photographed with CCD Olympus DP50.

### 2.7. Electron Microscopy

Scanning electron microscopy: Flies were anesthetized with medical ether and examined without fixation on a Hitachi TM-1000 machine.

Transmission electron microscopy (EM): Preparation and EM analysis of salivary gland polytene chromosomes were performed as described earlier [37], an electron microscope (JEM-100C) was used.

## 3. Results

### 3.1. Custom Deletions of the 5′ Region of the *Notch* Gene

First, we obtained the founder line *y w dN [w*–*]/FM0* (hereafter, dN/FM0), in which the regulatory (promoter) region of the gene, all known transcription initiation sites, and the first exon including the 5′UTR, the signal peptide sequence, and the subsequent nine amino acid residues, as well as about 1 kb of the first intron, were completely deleted(see Materials and Methods, Figure 1B–D). The Notch gene has no alternative promoters, and therefore, we have good reason to believe that the gene product is not produced, that is, dN [w–] is a null allele. Then we integrated a full-size fragment of the regulatory region of the gene (N-4kb) into the attP site of the founder line (see Materials and Methods, Figure 1D,E). As a result, three fly lines *y w N-resc[w+]* (further N-resc[w+]) were obtained, in which red-eyed homozygous females and hemizygous males were viable and fertile. The flies of these lines manifested a weak *Notch*-like phenotype: some males had small nicks on the wings, and partially rough facets. We asked whether this phenotype was attributable to the presence of auxiliary elements of the pGE-attB-GMR[N-4k] construct within the first intron of *Notch*. These auxiliary elements (denoted hereafter as AEs) included mini-*white* reporter gene under the control of heat-shock *hsp70* gene promoter and the GMR-enhancer, plasmid DNA, and *P*-element ends (Figure 1E). To address this question, the AEs were floxed out, and an N-resc[w–] line was established (Figure 1E), which had a normal phenotype. Thus, the 4-kb fragment of the construct rescued the lethality of the founder deletion and, therefore, contained the functional regulatory sequences of *Notch*.

Next, we obtained deletions removing various fragments upstream of the 5′UTR of *Notch* in the pGE-attB-GMR[N-4k] construct (see Materials and Methods, Figure 1F, Appendix A). When selecting fragments for deletions, we relied on the known data on the possible functional potential of different sites in this region. Thus, deletions d1, d2, and d3 are located in the region that exhibits insulator properties according to [21]. Deletion d2 removes 126 bp that fully include a 47-bp tandem repeat that was shown to be functionally important [17]. Deletion d3 removes a fragment of 255 bp, which is necessary for the interband formation in salivary gland polytene chromosomes de novo in the ectopic environment [38]. Deletion d1, in addition to the d2 sequence, removes 146 bp distal to the gene start. Deletion d4 spans the region of the assumed topological domain boundary (according to [29,30,31]). Nucleotide sequences of the deletions are shown in the Appendix A. Using *ΦC*31-mediated attP/attB integration, *Drosophila* lines were obtained that carry the four deletions described above, and the following homozygous stocks were established: *y w* d1[w+], *y w* d2[w+], *y w* d3[w+], and *y w* d4[w+] (furtherd1[w+]; d2[w+]; d3[w+]; d4[w+]). These were subsequently converted into white-eyed fly stocks d1[w–]; d2[w–]; d3[w–], and d4[w–] by crossing to the source of CRE and removal of AEs. Thus, these fly stocks harbor targeted deletions in the endogenous *Notch* locus.

### 3.2. Flies with Homozygous Deletions Display Mutant Phenotypes Only in the Presence of the AEs in the First Intron of *Notch*

We analyzed the phenotypes of the fly stocks obtained. Flies in all stocks with deletions, but without AEs, did not show any abnormalities in the gross morphology of eyes, wings or other organs, whose development is controlled by the *Notch* gene. Therefore, these deletions have no noticeable effect on the gene activity.

However, in the stocks where AEs were present, a range of eye phenotypes were readily observed. These phenotypes were temperature-dependent, and were stronger at 18 °C, than at 25 °C (Figure 2). In addition, eye phenotypes were more severe in males than in females. In the N-resc[w+], d2[w+], d4[w+] lines grown at 18 °C, the eyes of the males were nearly normal. Males of the d1[w+] line at 18 °C showed noticeable irregularities in the structure and location of the facets, while at 25 °C they had normal eyes (Figure 2). At 18 °C, males of the d3[w+] line had the strongest degree of eye phenotype. Namely, the facets were overall smaller, with interstitial tissue filling the space in between; the bristles, instead of their normal position in the facet corners, were often situated in the space between the facets or were missing altogether (Figure 2). In females of the d3[w+] line at 18 °C, the eye phenotype was less pronounced. At 25 °C, the males of the d3[w+] line had rough eyes, while the females had normal eyes.

Thus, the differences between the lines with deletions appear only when AEs are present in the first intron of the *Notch* gene, and these differences affect the eye morphology. Deletion d2 did not lead to any changes in the eye morphology, even in the presence of AEs, while the deletions d1[w+] and d3[w+] significantly disturbed the eye structure. The greatest effect on the eye morphology, similar to the effect of the “classic” *N^fa-swb^* deletion, was observed for the d3[w+] deletion (Figure 2, Appendix A). In addition, in this line, the viability of males was reduced (data not shown).Since we observe a stronger mutant phenotype (rough eyes) in the d1-d3 [w +] lines compared to N-resc [w +] (Figure 2, Appendix A), we believe that they are indeed more severely affected.

Interestingly, we found one more effect: in lines with deletions and AEs, there were some females and males with highly reduced external genital organs and underdeveloped ovaries and testes. In the founder line dN[w–]/FMO, as well as in the lines with deletions but without AEs, we did not detect such flies. However, in the lines with deletions and AEs, the frequency of such flies without genitals ranged from 2% to 7%, which was significantly different from the frequency of such flies in the N-resc[w+] line, which contains only the AE insert (less than 1%). Regardless of whether the flies were raised at 18 °C or 25 °C, there were no significant differences in the frequency of such flies. Previously, a *Notch* mutation (*fa-notchoid*) causing a range of phenotypes in male genitals was described in literature (0.05−5%) [39]. However, the events of complete genital reduction in both sexes associated with the mutations in the 5′ regulatory region of the gene, appear to be reported in our study for the first time.

### 3.3. In the Absence of a Normal Copy of the *Notch* Gene the Deletions with AEs Demonstrate Multiple Phenotype Disorders

To test the effect of the resulting deletions in the absence of a normal copy of the *Notch* gene, we crossed the dN[w–]/FMO founder females with the males of lines containing these deletions and analyzed the phenotypes of transheterozygous females. In lines with a chromosome without AEs (wild type OR, N-resc[w–], d1-4[w–]), transheterozygous females were viable, had normal eyes, sometimes small nicks on the wings, and poorly expressed delta veins (like in the original line dN[w–]/FMO).

Females that had their *Notch* locus containing AEs (N-resc[w+], d1-4[w+]) and were transheterozygous with the dN[w–] deletion, demonstrated a pleiotropic effect. Such females were sterile, had low viability and multiple morphological abnormalities: rough eyes, extra-thoracic bristles, abnormal wing structure and shape (Table 1, Figure 3). At 18 °C, the survival rate of trans-heterozygous females was significantly reduced in all lines containing AEs (data not shown).

In various deletions combined with AEs, the degree of manifestation of most phenotypic features was nearly equal except for the eye phenotype. The severity of phenotype in facet structure varied among different deletions (Figure 4). Whereas in the N-resc[w+]/dN[w–] and d4[w+]/dN[w–] females there was only a slight disorganization of the regular facet arrangement, in the d2[w+]/dN[w–] females the abnormalities were more pronounced. The most significant changes were observed in the d1[w+]/dN[w–] and d3[w+]/dN[w–] females: their facets were separated by interstitial tissue and displayed severe irregularities in the location of the eye bristles (Figure 4).

As in the case of homozygous lines, the temperature noticeably influenced the eye phenotype of trans-heterozygotes. At 18 °C, in N-resc[w+]/dN[w–] and d4[w+ ]/dN[w–] females, the abnormalities became much more noticeable, as “apoptotic” facets appeared; in d2[w+ ]/dN[w–] females some space appeared between the facets. In d1[w+]/dN[w–] females, the eye phenotype did not depend on the temperature. Females of the d3[w+]/dN[w–] line at 25 °C appeared with a very low frequency; the facet structure problems were the most severe. At 18 °C, such females showed almost negligible survival (ND in Figure 4).

Trans-heterozygous *N^fa-swb^*/dN[w–] females showed similar deviations: they had rough eyes, thickened delta wing veins, and nicked wings. However, tissue-specific differences from resc-dN[w+]/dN[w–] transheterozygotes were also observed. *N^fa-swb^*/dN[w–] females were fertile, and their triplo-array of bristles along the wing margin was not disturbed; at the same time they had an excessive number of bristles on the thorax and legs (hairy phenotype) and curved tibia of hind legs (Figure 3).

Our data indicate that in animals trans-heterozygous with the founder deletion, the presence of AEs in the first intron of *Notch* leads to the phenotypic defects if AEs are located *in cis* to the d1-4[w+] deletions. In case of AEs *in trans* position, as in N-resc[w–]/dN[w+] and d1-4[w–]/dN[w+] females, the phenotypes are normal (wild type). N-resc[w+]/dN[w+] and d1-4[w+]/dN[w+] females showed the same pleiotropic phenotypess as described above.

Thus, the effect of AEs on the *Notch* gene expression was observed on the background of the founder deletion and had a pleiotropic effect. Multiple phenotypic defects in transheterozygotes were more prominent at low temperatures, as well as in the combination with the d3 deletion.

### 3.4. Modeling the fa-swb Deletion

The X-ray-induced mutation *N^fa-swb^* [40] is a deletion of ~880 bp distal to the transcription initiation site ([17], Appendix A). This is a recessive hypomorphic allele, which leads to the rough eye phenotype in males; in females the eyes are nearly normal. In early studies it was shown that the manifestation of *N^fa-swb^* can be either suppressed or enhanced by various chromosomal rearrangements located both *in cis* and quite remotely [15,19,20]. The founder deletion of the entire regulatory region of the *Notch* gene, including the promoter, 5′-UTR, and part of the first intron, which we obtained here for the first time, allowed us to identify a strong pleiotropic effect in transheterozygous *N^fa-swb^*/dN females. Not only the eye morphology was disturbed, but also the formation of wings, bristles, and legs. The phenotype was similar to the N-resc[w+]/dN transheterozygotes comprising the AE insertion in the first intron, however there were some tissue-specific differences. We decided to recapitulate the *N^fa-swb^* deletion in the pGE-attB-GMR[N-4k] construct in order to perform a fair comparison.

First, the dfa-swb deletion was constructed (see Materials and Methods). Its nucleotide sequence was slightly different from the sequence of the *N^fa-swb^* deletion described earlier [17]: seven nucleotides were absent from the distal part of the dfa-swb deletion, and there were several nucleotide substitutions at the ends (Appendix A). Several independent dfa-swb[w+] lines were obtained. In all cases, homozygous deletions were lethal. After excision of the AE fragment flanked by loxP sites, homozygotes in the dfa-swb[w-] line also failed to survive. Using the balancer FM7, Tb, containing a larval marker, we found that animals carrying the homozygous deletion dfa-swb[w+] die at early stages of development, and homozygotes dfa-swb [w–] live to the pupal stage.

In order to dissect the possible important differences in DNA sequence between the classic *N^fa-swb^* deletion [40] and dfa-swb, we used PCR to obtain a *Cla*I-*Xho*I genomic DNA fragment (Figure 1D) from the *w^a^N^fa-swb^* line and replaced an analogous full-sized DNA fragment in the pGE-attB-GMR[N-4k] vector by this PCR product. As a result, we got an exact copy of the *N^fa-swb^* deletion in our system (further dfa-swb^LK^). Several independent fly lines were obtained. The dfa-swb^LK^[w+] homozygotes died at early stages of development, as did the dfa-swb[w+].However, the dfa-swb^LK^[w–] homozygotes survived and had a rough eye phenotype, similar to the animals with the classic *N^fa-swb^* deletion.

We conducted a study of dfa-swb allelic relationships with some of the deletions. The dfa-swb[w-]/*N^fa-swb^* females had the rough eye phenotype, and the dfa-swb[w+]/*N^fa-swb^* transheterozygotes additionally had defects in wing and bristle formation, which is apparently due to the AE insertion. The dfa-swb[w+]/N-resc[w+] flies exhibited a pleiotropic phenotype similar to the N-resc[w+]/dN transheterozygotes (thickened delta veins, nicked wing edge), and the dfa-swb[w+]/N-resc[w–] females had a normal phenotype. Non-reciprocal allelic relationships were observed: dfa-swb[w+]/d3[w–] females had a normal phenotype, and dfa-swb[w–]/d3[w+] females had rough eyes.

### 3.5. Cytological Analysis of Mutations in the 5′ Region of the *Notch* Gene

In this study, we identified DNA sequences in the 5′ end of the *Notch* gene, whose removal in combination with the insertion of the auxiliary transgene elements into the first intron changes the gene activity. We proceeded to analyze how these genetic changes affected the chromatin structure in the region of this gene localization. Larval salivary gland polytene chromosomes represent the most convenient test system for such analysis, since direct visualization of the interphase chromatin structure is possible. Previously, using high-resolution fluorescent in situ hybridization (FISH) on salivary gland polytene chromosomes, it was shown that the 5′-flanking region of the *Notch* gene is located in the 3C6/C7 interband, and the gene body is almost completely located in the 3C7 band [16]. This is consistent with cytological data on *N^fa-swb^* deletion, which affects a substantial part of the 5′-untranscribed gene region and causes the disappearance of the 3C6/C7 interband with fusion of the 3C5-6 and C7 bands [14,41] and Figure 6E)

According to Bridges’ map, in the region of the *Notch* gene localization there are two thick bands 3C2-3 and 3C5-6, and two thin bands 3C1 and 3C7 (Figure 6A). In the majority of lines with AE insertion (N-resc[w+], d1[w+], d2[w+], d4[w+]) when raising flies and making chromosome preparations at a stable 25 °C temperature, we observed an enlargement of the decondensed region in the 3C6/C7 interband (e.g., see Figure 5A,B, line N-resc[w+], black arrow), bordered at its proximal end by a thin gray band, which we attribute to the band 3C7. After the heat shock in all lines with AEs in this area, a new puff is formed (e.g., see Figure 5B, line N-resc[w+], square bracket). One of the most obvious reasons for the formation of a new puff in the lines containing AEs is the presence of the functional promoter of the heat shock (HS) gene *hsp70* in the transgene. Data on FISH localization of the full-sized transgenic DNA pGX-attP[3′-5′ HA-N] probe in the 3C region of the N-resc[w+] line indicate that the artifactual decondensed region (in the absence of HS) and the puff (when exposed to HS) are entirely formed by the transgene material (Figure 5B).

For mapping deletions, lines were analyzed after removal of AEs at the lox P sites by Cre recombinase (Figure 1E,F). In N-resc[w–] and d4[w–] lines the banding pattern in 3C region reverted back to normal (Figure 6B). In d1[w–], d2[w–], and d3[w–] lines, as well as in *N^fa-swb^,* the 3C6/C7 interband collapsed and the 3C5-6 and 3C7 bands fused. (Figure 6B–D, respectively). After HS, puffs did not form in any of these lines (data not shown).

A slightly different chromosome structure in region 3C was found in the d3[w+] line, and in in the dfa-swb[w+]/FM0 and dfa-swb^LK^[w+]/FM0 lines. In spite of the presence of the insertion giving artifactual expansion of the interband in other lines with deletions, we consistently detected the “collapse” of the 3C6/C7 interband, as is typical for the “classical” *N^fa-swb^* deletion (Figure 5A).

In these lines this may be due to the removal of the functional elements in the promoter region of the *Notch* gene necessary for the formation and maintenance of “open” chromatin state in this region and regulation of this gene (for example, binding sites of insulator proteins, remodeling proteins, DHS sites, etc.).

In order to check whether the deletions obtained remove the region of the putative insulator [21], we performed polytene chromosome immunostaining with antibodies against CHROMATOR and CTCF insulator proteins [42].It turned out that in contrast to the wild-type control X chromosomes, no signals of these proteins are observed in the 3C6/C7 region of d3 and N^fa-swb^ chromosomes, while in other lines the localization patterns were unaltered (Figure 7).

Thus, it can be assumed that d3 and N^fa-swb^ deletions directly or indirectly disrupt the binding of CTCF and CHROMATOR insulator proteins in 3C region, and this may cause incorrect regulation of the *Notch* gene due to changes of chromatin topology in a given chromosome region, for example, through the destruction of TAD borders or enhancer–promoter interactions.

## 4. Discussion

In this work, we conducted targeted editing of the *Notch* gene 5′ region in its native environment using CRISPR/Cas9-induced homology-directed repair system combined with the site-specific recombination method, and studied the genetic and cytological effects of these changes. We generated a founder fly line, in which the 4-kb fragment containing the promoter region, the first exon, and a part of the first intron of the *Notch* gene was replaced by anattP site. Then, we reintegrated the full-size fragment of the 5′ region of the gene into the attP site and restored the functionality of the regulatory sequence. Insertion of AEs in the first intron of *Notch* gene behaved as a hypomorphic mutation. In the homozygous state, it caused a weak *Notch*-like phenotype: small nicks on the wings in males and partially rough eyes. In trans-heterozygote with the founder deletion it caused multiple phenotypic defects in the *Notch* target tissues. One way to explain the mutant phenotype associated with AE insertion is that fewer normal transcripts are being produced. The AE insert contains a mini-white gene with a transcription terminator in its 3′ UTR, which may result in premature transcription termination. This effect was previously reported for the mutation caused by the insertion of the P{EP} transposon into the 5′-noncoding region of the Trl gene [43]. According to our preliminary data obtained using RT-PCR in resc [w+] oocytes the number of normal transcripts was reduced three times. 

Another possible explanation is based on aberrant splicing, which leads to the formation of chimeric RNA and lower levels of normal transcripts. However, in our case, this seems unlikely, since the sequence of the mini-white reporter does not contain introns and in experiments with cDNA we did not reveal any altered transcripts (preliminary data). Due to the presence of this insertion, a set of six deletions that overlapped most of the 5′ regulatory region of the gene allowed us to identify DNA segments involved in chromatin organization and regulation of the *Notch* gene.

The summary table of phenotypes of deletions is presented in Appendix A. In the lines without AEs, the deletions d1–d4 did not display any phenotypic abnormalities in the structure of the eyes, wings, or other organs controlled by the *Notch* gene during development, which indicates the absence of a noticeable effect of these deletions on the gene activity. The lines that contained the same deletions in combination with AEs in the first intron showed mutant phenotypes. Most of the deletions we generated in combination with the AE insertion affected the eyes. Defects of the eye structure in the lines d1[w+] and d2[w+] were mild. Maximum defects of the eye phenotype were observed in the d3[w +] line—flies had rough eyes, as in the line with the classic *Notch* allele, *N^fa-swb^*. In earlier transgenesis experiments it was shown that the DNA fragment, including the sequence deleted by *N^fa-swb^*, had insulator properties. It protects reporter transgenes from the position effect and blocks the enhancer–promoter interactions [21]. Based on the transgenic analysis of DNA fragments from the region of the *Notch* locus, the authors suggested that the sequences important for insulator activity are located within the region that includes the *N^fa-swb^* sequences and 60 bp closer to the proximal breakpoint of this deletion. A study of chromosome structure and chromatin organization in this area showed the presence of several closely located nuclease hypersensitive sites (Figure 8), which are also located in the proximal part of the *N^fa-swb^* sequence and in the promoter region of the *Notch* gene [21]. ModEncode data analysis [44] revealed a significant local decrease in nucleosome density in *Drosophila* cell cultures in the 5′ region of the *Notch* gene. In this region, pronounced binding peaks of a number of insulator proteins (CP190, CTCF, and CHROMATOR (Chriz)) are detected, with the tops of the peaks in all these cases coinciding with the position of the DNA fragment capable of forming an interband [28,38], (Figure 8). In this context, it should be mentioned that a significant fraction of insulator proteins have been demonstrated to map to the boundaries of topologically associated domains, and these boundaries are cytologically related to the polytene chromosome interbands [31,45].

We previously showed [38,47] that three DNA fragments (1504 bp, 914 dp, and 276 bp) from 5′ nontranscribed region of the *Notch* gene, being transferred in a new genetic environment (region 84F, 3R chromosome) give rise to a novel interband. After removing 246 bp from the 1504-bp DNA fragment the formation of a new interband does not occur. Thus, it was concluded that ~250 bp-long proximally located DNA subfragment from the *N^fa-swb^* deletion region is necessary and sufficient to form the interband in an ectopic environment. Based on this, we constructed the deletions d1, d2, and d3 so that they consistently overlap the given region of 250 bp, which also demonstrates the insulator properties (Figure 8B). It turned out that only the d3 deletion, according to our immunolocalization data, removes the binding sites of the insulator proteins CTCF and CHRIZ (Figure 7), which is an additional argument in favor of the fact that precisely this sequence serves as an insulator and is necessary for interband formation.

According to our cytological data, the 3C6/7 interband disappears in the d1[w–], d2[w–] and d3[w–] lines, but in combination with AE insertion, the interband disappears only in the d3[w+] line. It may be due to the removal of the functional elements in the promoter region of the *Notch* gene that are necessary for the formation and maintenance of open chromatin state in this region as well as for the regulation of the *Notch* gene as a whole, for example by changing the chromatin topology in a given chromosome region, or by breaking TAD boundaries or enhancer–promoter associations. It has previously been suggested that there is an interaction between enhancers located in the introns of the Notch gene and regulatory elements of the promoter zone [17,21]. Recently, using the Hi-C method [31], a physical interaction was shown between the promoter region of the Notch gene and the sequence in the body of this gene with TAD formation of about 20 kb. It can be assumed that if AE is inserted into the first intron, physical interactions between the hypothetical enhancer and the insulator sequence in the promoter region of the gene are disrupted. Further experiments using the Hi-C method can help test this hypothesis using the deletions we obtained.

A change in chromatin structure in the region of 3C6-7 and phenotypic disorders in mutants obtained may also be due to the fact that the GMR enhancer embedded in AE might competitively interact with the promoter of the *Notch* gene in the absence of an insulator sequence. To test this hypothesis, we obtained the N-resc-dGMR[w+] line, which comprised the AE insertion without the GMR enhancer, and the d3-dGMR[w+] line, in which the insertion without the GMR enhancer was combined with the d3 deletion. Phenotypic manifestations in both homozygous flies in these lines and in trans-heterozygotes with a founder deletion were the same as in the original N-resc[w+] and d3[w+] lines (data not shown). This indicates that the GMR enhancer does not seem to significantly affect the interaction between the AE insertion and the promoter region of the gene. The fact that the mutant phenotypes in the lines containing the AE insertion were temperature-sensitive indirectly supports the idea that competition can occur between the promoter of the *Notch* gene and the *hsp70* promoter of the auxiliary element, whose activity is likewise temperature-dependent.

In early genetic studies, allele interactions of the *Notch* gene-recessive mutations of the *fa* group (*facet* (*fa*)*; facet-glossy* (*fa-g*)*; split* (*spl*)) with the *N^fa-swb^* mutation were described. Mutation s*facet(fa)* affect eye morphology and are associated with the insertions of transposable elements in the second intron of *Notch* [18]. In combination with *N^fa-swb^*, these mutations exhibited an enhanced phenotype: in *N^fa-swb^fa^g^* males, the wing veins were thickened and deltas were formed at the wing margins. There was enhancement of the hair distribution on the thorax and legs of *N^fa-swb^spl* flies [48]. This suggests that there are some regulatory elements in the introns of the *Notch* gene that interact with the functional sequences of the promoter region.

It was previously shown that the phenotype of the “classic” deletion *N^fa-swb^* can either be enhanced or suppressed in the presence of rearrangements located rather distal [15,19,20]. In the work by [48], a spontaneous variant of *N^fa-swb^* referred to as *facet-strawberry-hairy* (*fa-swb^h^*) was detected. The homozygotes and hemizygotes demonstrated much more severe mutant phenotype manifestation than *facet-strawberry* animals. In these flies, the eyes were slightly narrow, very rough, and glossy. There was a proliferation of microchaetae over the thorax, abdomen, and legs, giving the animals a “hairy” appearance, and the tibiae of the hind legs were distinctly bowed. Under the assumption of Welshons and Welshons [20] the *facet-strawberry-hairy* phenotype was caused by the interaction of *N^fa-swb^*with an “enhancer” that is closely linked to *Notch* and distal to it. The nature of this factor is still unknown. In our work, a similar pleiotropic phenotype was demonstrated by the *N^fa-swb^*/dN trans-heterozygous females, and, apparently, it can be assumed that if such a regulatory factor exists, it is located within the region removed by the founder deletion, as rare surviving *N^fa-swb^*/dN females essentially mimicked the *fa-swb-hairy* phenotype described by [48]. 

When we generated a dfa-swb^LK^ deletion that completely copied the structure of the *N^fa-swb^* deletion, it turned out that in the dfa-swb^LK^[w+] line, where the deletion was combined with the AE insertion, the homozygotes did not survive, but in the dfa-swb^LK^[w–] line the flies were viable and had rough eyes similar to the *N^fa-swb^* phenotype. Thus, the presence of the AE insertion led to the phenotype enhancement, as it was observed in the lines with other deletions that we obtained. Thus, the AE insertion into the first intron itself produced a mutant phenotype, and significantly interfered with gene activity in combination with deletions affecting the functional elements of the regulatory region.

Of particular interest is the fact that our dfa-swb deletion, which completely removes the *N^fa-swb^* deletion sequence, was lethal and showed complex allelic interactions with mutations in the 5′ regulatory region of the gene. There were minor differences in the dfa-swb sequence as compared to dfa-swb^LK^. Namely, seven nucleotides were missing from the distal part of the deletion, and several nucleotides at the ends flanking the deletion were replaced (Appendix A). According to Kidd [49], the CAT-box motif necessary for RNA polymerase II binding is present in the region of the *Notch* gene core promoter. In the work by Ramos and colleagues [17], this element sequence was hypothesized to be disrupted at the proximal breakpoint of the *N^fa-swb^* deletion, which led to altered gene expression and the appearance of the *N^fa-swb^* phenotype. In our experiments, we obtained several deletion variants that also affected this area. In the line with the dfa-swb deletion (Appendix A), the expected CAT-box sequence is significantly different from that in the dfa-swb^LK^ and *N^fa-swb^* lines, which apparently causes a significant transcription defect and leads to a lethal phenotype. At the same time, the d3 sequence also features certain sequence changes in this region (Appendix A); however, the flies in the d3[w–] line look normal. Whether these differences are functionally important, or the lethality of a dfa-swb deletion is due to other reasons, requires further investigation.

Notably, we found paradoxical allelic interactions between the dfa-swb deletion and the d3 deletion. It turned out that the phenotype of trans-heterozygous females depends on the deletions carrying the insertion of AEs. If AE is placed *in cis* with respect to d3, then the trans-heterozygotes dfa-swb[w–]/d3[w+] have coarse eyes, and if the AE is on the same chromosome with dfa-swb, then the eye phenotype of females dfa-swb[w+]/d3[w–] is normal. Such non-reciprocal allelic interactions likely indicate that communication between the putative regulatory elements of the first intron and the functional sites of the promoter region is broken. Deciphering the mechanism of these intricate interactions will be the subject of further research.

## Figures and Tables

**Figure 1 genes-10-01037-f001:**
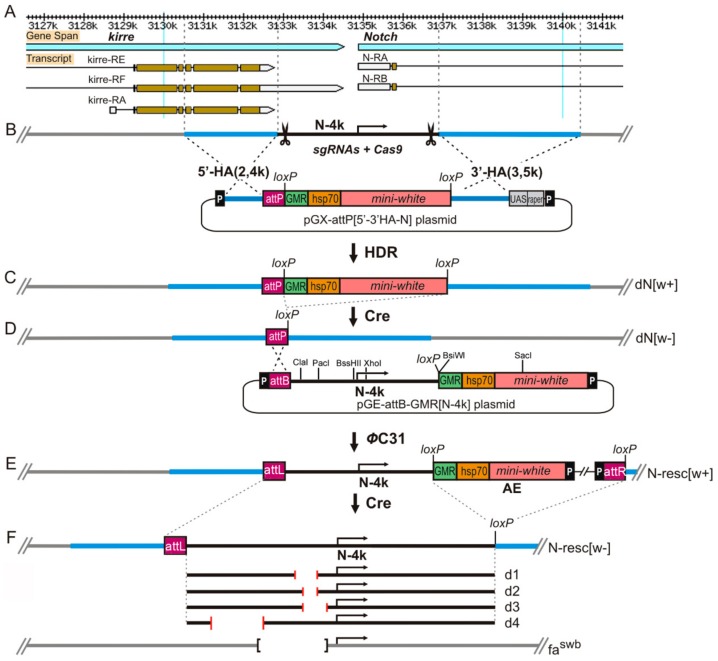
CRISPR/Cas9-induced homology-directed repair (HDR) with a pGX-attP{3′-5′ HA-N} donor vector. (**A**) Molecular and genetic map of the *Notch* locus. (**B**) Scheme of the 5′ region of *Notch* gene and HDR strategy utilized to replace the region with attachment Phage (attP) landing site of *ΦC*31 phage integration system. The target sites are indicated as “scissors”. Homology arms of 2.4 kb and 3.5 kb immediately flanking the cleavage sites were cloned into pGX-attP vector. The pGX-attP[3′-5′ HA-N] vector contains an attP site for subsequent access to a targeted locus and a mini-*white* gene under control of the *hsp70* promoter and GMR (gene modulating RNase II) enhancer, which drives expression in the eye, flanked by *loxP* recombination sites for its removal. (**C**) The pGX-attP[3′-5′ HA-N] donor and the plasmids encoding guide-RNAs (gRNAs) were injected into vasa-Cas9 embryos for generation of the w+ founder line (dN w+). Mini-white cassette was then removed by Cre and a founder line carrying only single attP site was created (dNw–). (**D**) A full-size 4-kb fragment of the 5′ region of the *Notch* gene was cloned into the pGE-attB-GMR [w+] vector, which was then integrated into the deletion region of the founder line through *ΦC*31-mediated DNA integration. (**E**) As a result of integration, the target region was restored at its original genomic locus together with w+ and vector sequences (AEs). Extra vector sequences, together with w+, were removed by Cre recombinase to produce the fly stock where in the engineered target region was flanked by attR and *loxP* sites (Nresc [w–]). (**F**) Schematic diagram of deletions in the 5′ region of the *Notch* gene (more details are given in the text).

**Figure 2 genes-10-01037-f002:**
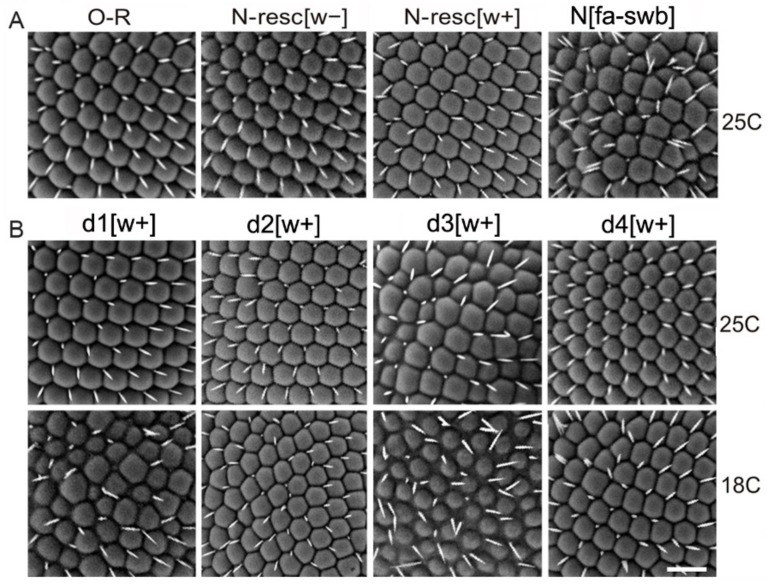
Scanning electron micrographs of the male eyes hemizygous for the indicated genotypes. Flies carrying deletions and AE [w+] were raised at 25 °C and 18 °C. The eyes of N-resc[w+] and N-resc[w–] males display minor abnormalities of the bristle and ommatidial pattern. The phenotypes of d3[w+] and fa-swb are more extreme. For more details see the text.

**Figure 3 genes-10-01037-f003:**
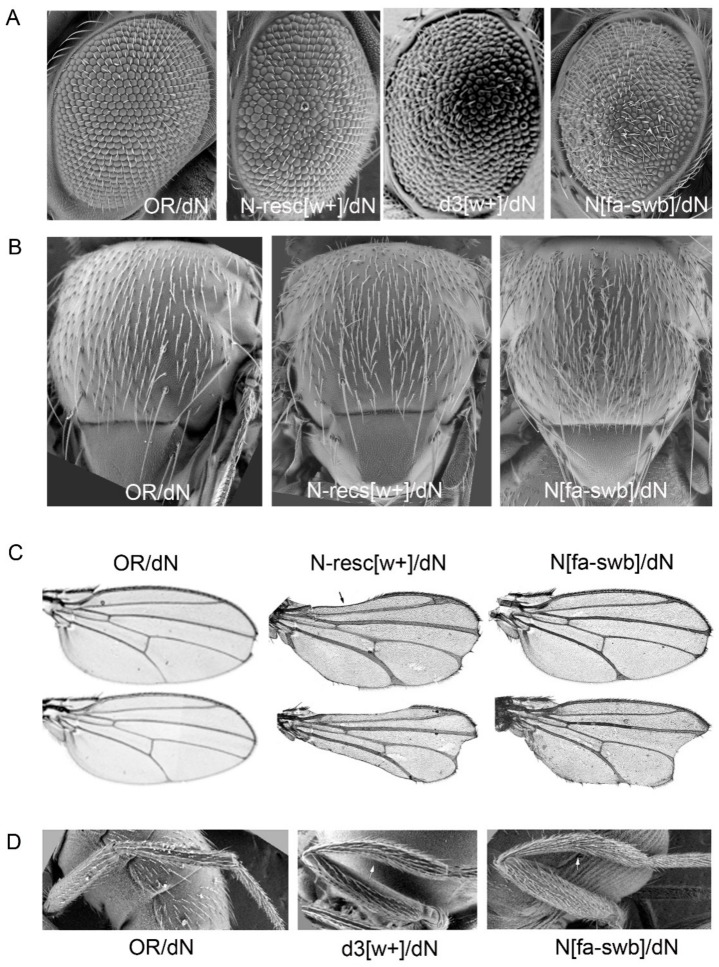
Pleiotropic effects in females trans-heterozygous for the founder deletion dN. (**A**) Defects in the facet structure. (**B**) There is an enhancement of the hair distribution on the thorax and legs of fa-swb/dN flies (“hairy” phenotype). (**C**) The wing veins are thickened, deltas are formed at the wing margins and apical notches are seen. The N-resc[w+]/dN females have gaps of the wing margin. The range of the phenotypes obtained are shown. (**D**) Curved tibia.

**Figure 4 genes-10-01037-f004:**
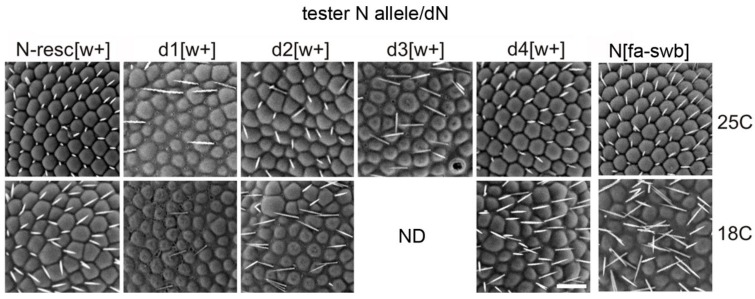
Scanning electron micrographs of the eyes of females trans-heterozygous for the founder deletion chromosome dN, raised at 25 °C and 18 °C. For description see the text.

**Figure 5 genes-10-01037-f005:**
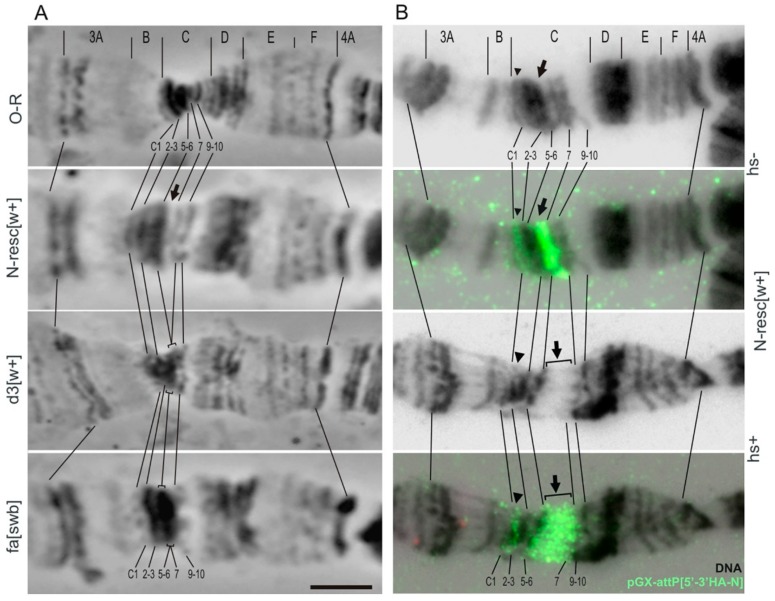
Truncation of the *Notch* 5′ end region changes the cytological structure of the 3C region in the polytene X chromosome. (**A**) Phase contrast images: OR (wild-type fly line, control), N-resc[w+] and d3[w+] (targeted mutations in the 5′ end region of *Notch* gene), and fa[swb] (*N^fa-swb^* deficiency in the 5′ flanking region of *Notch*). (**B**) Fluorescent in situ hybridization (FISH) localization of the DNA probe pGX-attP{3′-5′ HA-N} (green) in the 3C region of the N-resc[w+] line. DNA staining (black).Arrows point to hybridization signal of the probe pGX-attP[3′-5′HA-N] in the artificial (thicker) interband (in normal conditions, hs-) or in the artificial puff (after heat shock, hs+), marked with a bracket. Arrowheads indicate additional signal in the 3C1–C2 region, where the *white* gene is situated. Cytological map of the 3C region according to Bridges (1938) is shown on top of each panel, lines connect some of the marker bands. O-R: wild-type line; fa-[swb]:*N^fa-swb^*deletion. Bar, 5 µm.

**Figure 6 genes-10-01037-f006:**
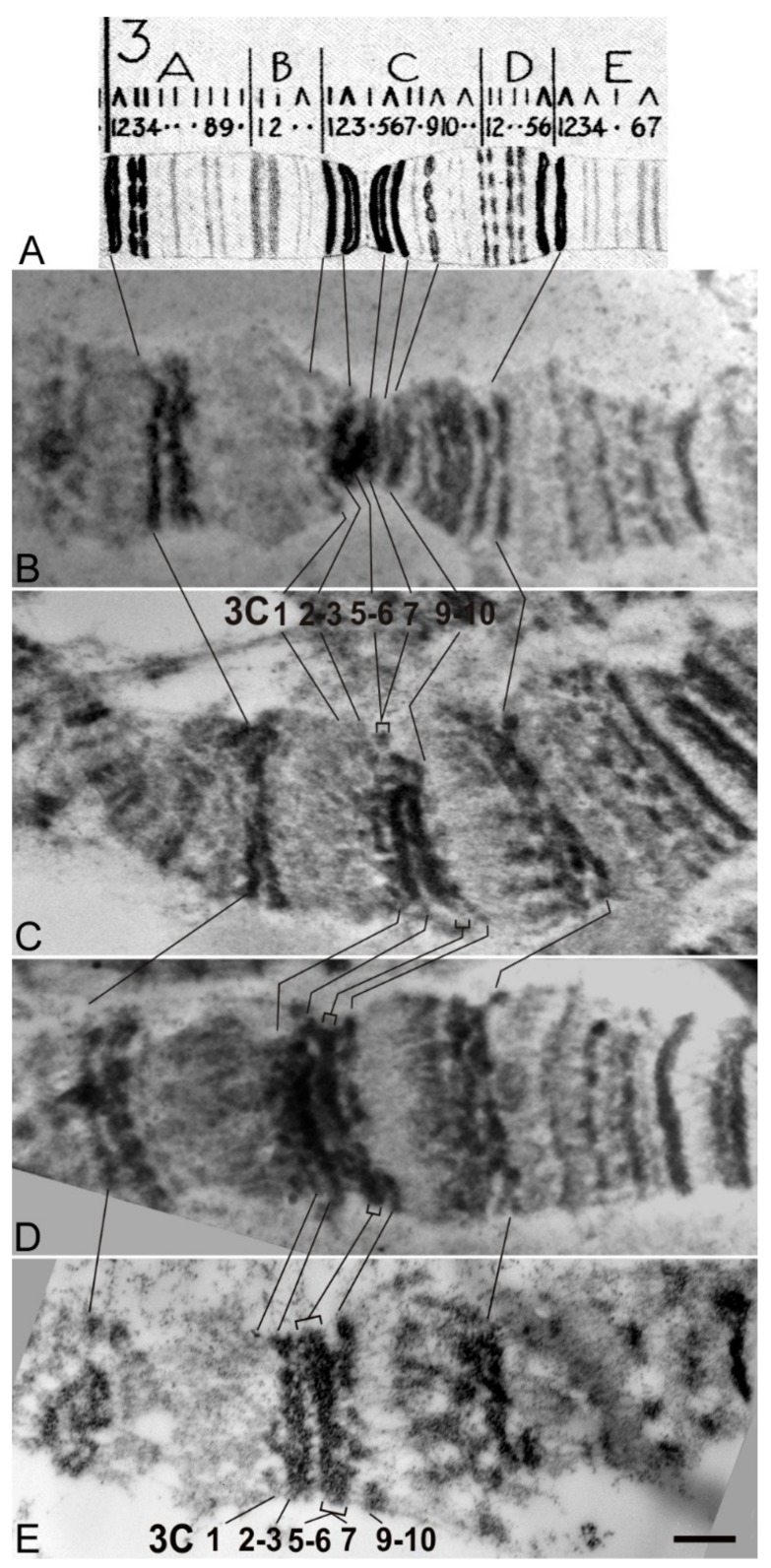
The 3C region of the polytene X-chromosome. (**A**) Bridges’ map, (**B**) Chromosomes from the N-rescue[w–], (**C**) d1[w–], (**D**) d3[w–], and (**E**) N^fa-swb^ flies. Bar, 1 µm.

**Figure 7 genes-10-01037-f007:**
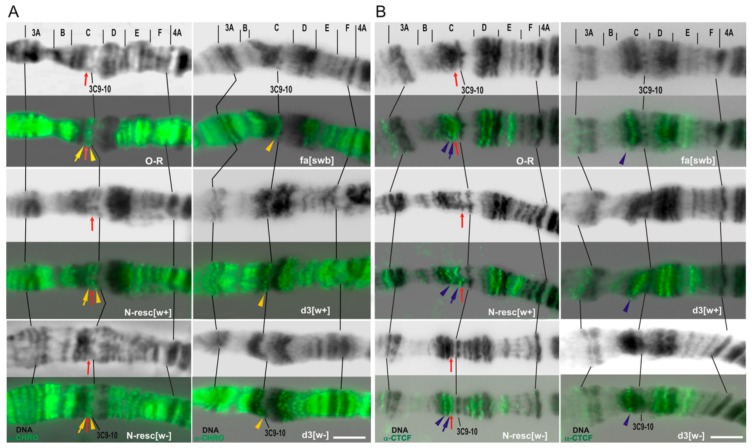
Indirect immunostaining of insulator proteins CHROMATOR (**A**) and dCTC-Factor (**B**) in the 3C region of the polytene X chromosome. DNA staining (black), CHROMATOR (CHRO) and CTCF signals (green). Red arrow*s* point to the thin band 3C7, containing the body of the *Notch* gene. The top in each panel features the cytological map of 3C region according to Bridges (1938), lines connect some of the marker bands. Yellow and blue arrows show an overlay of CHRO and dCTCF immunostaining signals in 3C6/C7 interband, respectively. Yellow and blue arrowheads show CHRO and dCTCF signals adjacent to the above signals and used as a reference. O-R: wild-type line; fa-[swb]:*N^fa-swb^* deletion. Bar, 5 µm.

**Figure 8 genes-10-01037-f008:**
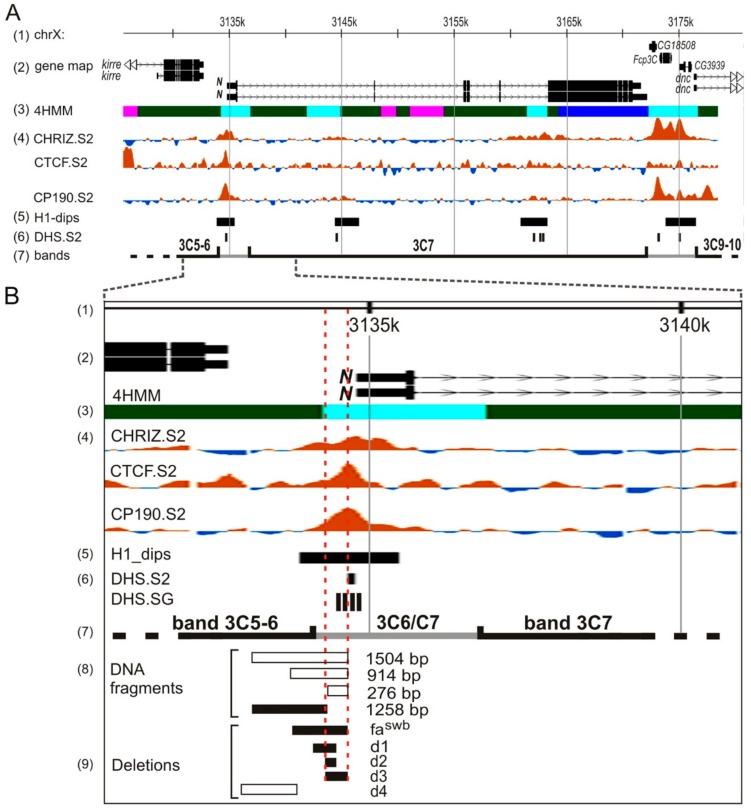
Map of the *Notch* locus (**A**) and 5′-end of the *Notch* gene on a larger scale (**B**). (1) Genomic coordinates (kbp). (2) Gene map. (3)4HMM-derived chromatin map [46]. (4) Enrichment profiles of insulator proteins in S2 cells (modENCODE project, http://intermine.modencode.org). (5) Histone H1 dip localization in Kc cells [25]. (6) DNase I hypersensitivity sites (DHS) in S2 cells [27] and in salivary glands of third-instar larvae [22]. (7) Borders of the bands in the region, as defined by the 4HMM. (8) DNA fragments capable (white) and not capable (black) of forming an interband in a new genetic environment [38,45]. (9) Deletions of DNA segments from the 5′-regulatory part of the *Notch* gene causing (black) and not causing (white) the collapse of the endogenous interband 3C6/C7 (current investigation).

**Table 1 genes-10-01037-t001:** Phenotypes of some heterozygous females tester N allele/dN[w-].

Tester N allele	dN[w–]				
	Viability(N allele/dNotch:N allele/FM0)	Eyes	Wings	Bristles	Legs
**rescue[w+]**	18:366	R	VTN	Ma	+
**rescue[w–]**	186:203	+	+	+	+
**d3 [w+]**	7:230	RG	VTN	Ma	C
**d3 [w–]**	183:175	R-	+	+	+
**N^fa-swb^**	42:425	R	VN	H, Ma	C

Notes: R: rough eyes; RG: rough and glossy eyes; H: phenotype "hairy" - extra and misaligned bristles on thorax and legs; Ma: additional bristles on the scutellum; V: thickened wing veins forming deltas at margin; T: gaps of triplo-row, N: nicked wings, C: curved tibia of hind legs; **+** indicates normal phenotype, **-** indicates the expression is slight or is a variable, overlapping wild type.

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
