# Peer review of "Structural and Functional Dissection of the 5′ Region of the Notch Gene in Drosophila melanogaster"

_genes, 2019, doi:10.3390/genes10121037_

Round 1
Reviewer 1 Report
Volkova and colleagues analysed the 5' regulatory region of the Notch locus in Drosophila. The study is based on the classical Notch allele fs-swb that disrupts a domain boundary. Rather elegantly they used CRISPR/Cas9 to replace a 4kb fragment including the 5' nontranscribed region , the first exon, and a part of the first intron with an attP site. They then integrated rescue constructs each with a small deletion in or close by the 5' regulatory region affected by fs-swb. The deletion themselves did not show any phenotype. However, the deletion in combination with an auxillary element further downstream in cis displayed Notch phenotypes. Based on further genetic experiments, the analysis of polytene chromosome banding pattern and immunofluorescent staining for insulator proteins the authors supported the notion that the 5' upstream regulatory region hosts an insulator element.
The study was conducted by using state of the art genetic methods and the data is displayed at high quality.
Minor comments:
1) It is not completely clear to me whether the deletion of the insulator element lead to a Notch loss- or gain-of-function. Did the authors looked at Notch protein levels by quantitative methods such as Western blot?
2) line 356: morphology
3) line547: intron
Author Response
Thank you very much for your constructive and useful comments. We tried our best to take your recommendations into account. This greatly improved the presentation of our data.
Minor comments:
1) It is not completely clear to me whether the deletion of the insulator element lead to a Notch loss- or gain-of-function. Did the authors looked at Notch protein levels by quantitative methods such as Western blot?
A: We believe that the mutation of the insulator element is a loss-of-function, since the mutant phenotype is enhanced in the heterozygote with a basic deletion of dN. Currently, we do not have the technical ability to carry out Western blot, but in the future we plan to do this as well as the measure the transcriptional activity of this mutant by q-PCR.2)
2) line 356: morphology
Done
3) line547: intron
Done
Reviewer 2 Report
In this study, the authors take a detailed approach to define regulatory elements of the Notch promoter in Drosophila melanogaster. As pointed out in the manuscript, the roles of this evolutionarily conserved signaling component in many different developmental contexts has been studied at the protein level, however, specific DNA segments important for expression of the gene remain to be characterized. Earlier studies highlighted the importance of an 880 bp segment 5’ to the transcriptional start site. Using an ingenious CRISPR-mediated re-engineering of the endogenous locus, the authors remove a segment spanning the TSS and replace it with a recombination site, allowing the rescue with either wild-type or modified promoter segments lacking key sequences. Interestingly, few or no phenotypes are observed with flies bearing various deletions unless an additional marker transgene (AE) is present in the first intron of N (the marker was preent to identify the correct recombinant stocks, and subsequently removed directed recombination. Using the “AE” containing sensitized alleles, the authors document wing, eye, bristle, and other developmental phenotypes that are in many cases temperature sensitive. In particular, deletion 3 has severe effects; this portion overlaps a mapped CTCF interaction area of the promoter. The authors test the effet of these mutations on chromatin structure by microscopic examination of polytene chromatin structure, and document that functional properties correspond with the banding patterns observed in the transgenic stocks. Loss of the CTCF/Chromator region corresponds to loss of binding of these proteins on polytene preparations.
Overall, this is an interesting study that provides further insight on classical N mutations that likely impacted expression of this gene. The use of CRISPR alleles in the endogenous allele is an ingenious test for the properties of specific proximal elements, some of which had previously been demonstrated to possess promoter-like (polytene band-forming) activities.
Specific questions for the authors are noted below:
The authors note that presence of AE in cis, but not in trans, produces the observed phenotypes, and speculate about possible promoter-enhancer interactions with the transgene elements. A direct model for how a transgene insert in the first intron would make a hypomorphic allele is that transcriptional readthrough is diminished by the 3’ UTR of the miniwhitegene (not shown on Fig. 1, but most likely present). The authors don’t mention this obvious possibility – have they looked at mRNA expression directly in their mutant lines? A second possibility to explain the cis effect of the AE insertions is that splicing is disrupted; again, have they tested this idea by measuring retention of intron 1 in mRNAs (improper splicing may itself lead to reduced mRNA stability). At the very least, the manuscript should present these two likely processes as possible explanations for the impact of AE on phenotypes. The work provides strong support for the role of promoter segments interacting with boundary element associated factors. Are there other regulatory factors likely interacting with these proximal promoter regions? (They cite old studies highlighting CAT sequences; what has been found since then?). What level of sequence conservation is found at the regions investigated here; are there conserved motifs for CTCF or other factors indicating that this regulatory structure is under selection? The paper was well written and for the most part very clear. A number of grammatical improvements/spelling corrections are noted below:
we defined a DNA segment…..[which when deleted leads to] which deletion leads to disappearance
[The] Notch pathway is an important and evolutionarily conserved signaling system
nervous system, [and] the development
[The] Notch signaling pathway regulates local interactions
between cells during [no “the”] the eye formation
Decreased Notch activity results in that differentiating retinal cells choose unusual development pathway, thereby leading
[Decreased Notchactivity causes differentiating retinal cells to choose an unusual development pathway]
In [the] D. melanogaster genome
Hundreds of mutations were [have been] obtained for the Notch locus
All mutations in this group disarrange [affect] the correct formation
Infuence
Cre and a w- fouder
eye morfology
First, [the] dfa-swb deletion was constructed
homozygotes in the dfa-swb[w-] line [also] failed to survive [no either] either.
animals carrying [the] homozygous deletion
untranscribed gene region, [and] causes the disappearance of the 3С6/C7 interband [with] fusion
by a thing gray band
After HS, puff[s] did not form
Artifact[ual] expansion of the interband
wilde-type [wild-type] control X
Author Response
Thank you very much for your constructive and useful comments. We tried our best to take your recommendations into account. This greatly improved the presentation of our data.
The authors note that presence of AE in cis, but not in trans, produces the observed phenotypes, and speculate about possible promoter-enhancer interactions with the transgene elements. A direct model for how a transgene insert in the first intron would make a hypomorphic allele is that transcriptional readthrough is diminished by the 3’ UTR of the miniwhite gene (not shown on Fig. 1, but most likely present). The authors don’t mention this obvious possibility – have they looked at mRNA expression directly in their mutant lines?A: We agree that one of the most likely explanations is premature transcription termination due to the transcription terminator of the mini-white gene. Similar effects have been reported for the insertion of the P{EP} transposon into the 5'-noncoding region of the Trl gene (Karagodin et al., 2016). According to our preliminary data obtained using RT-PCR in resc [w+] oocytes three times reduced the number of normal transcripts. However, since these data are preliminary, we do not refer to them in our article (changes made to discussion) line 480-490
A second possibility to explain the cis effect of the AE insertions is that splicing is disrupted; again, have they tested this idea by measuring retention of intron 1 in mRNAs (improper splicing may itself lead to reduced mRNA stability). At the very least, the manuscript should present these two likely processes as possible explanations for the impact of AE on phenotypes.A: In our case, this seems unlikely, since the sequence of the mini-white reporter does not contain introns and in experiments with cDNA we did not reveal any altered transcripts (preliminary data).
The work provides strong support for the role of promoter segments interacting with boundary element associated factors. Are there other regulatory factors likely interacting with these proximal promoter regions? (They cite old studies highlighting CAT sequences; what has been found since then?).A: The analysis of literature did not reveal any new works about the regulatory factors likely interacting with these proximal promoter regions this gene.
What level of sequence conservation is found at the regions investigated here; are there conserved motifs for CTCF or other factors indicating that this regulatory structure is under selection?
A: We tried to assess the level of conservatism in the study area by comparing DNA in all currently sequenced natural Drosophila. However, preliminary analysis did not reveal any specific conservative regions of the gene regulatory zone. This is an interesting area of research, and it requires additional serious bioinformatic analysis.5. The paper was well written and for the most part very clear. A number of grammatical improvements/spelling corrections are noted below:
Thank you very much, all grammatical improvements/spelling corrections have been made.
we defined a DNA segment…..[which when deleted leads to] which deletion leads to disappearance
[The] Notch pathway is an important and evolutionarily conserved signaling system
nervous system, [and] the development
[The] Notch signaling pathway regulates local interactions
between cells during [no “the”] the eye formation
Decreased Notch activity results in that differentiating retinal cells choose unusual development pathway, thereby leading
[Decreased Notch activity causes differentiating retinal cells to choose an unusual development pathway]
In [the] D. melanogaster genome
Hundreds of mutations were [have been] obtained for the Notch locus
All mutations in this group disarrange [affect] the correct formation
Infuence
Cre and a w- fouder
eye morfology
First, [the] dfa-swb deletion was constructed
homozygotes in the dfa-swb[w-] line [also] failed to survive [no either] either.
animals carrying [the] homozygous deletion
untranscribed gene region, [and] causes the disappearance of the 3С6/C7 interband [with] fusion
by a thing gray band
After HS, puff[s] did not form
Artifact[ual] expansion of the interband
wilde-type [wild-type] control X"
done
Reviewer 3 Report
Volkova et al used genome engineering technologies, based on CRISPR/Cas9 and phiC31 recombinase, to engineer precise deletions in the upstream region of the Notch gene. Previously only one allele (fa-swb) had been characterized in that region, which deleted a region shown to be an insulator and behaved as a hypomorph. Here, the authors generated six new deletions, d1-4, as well as two mimics of the fa-swb original deletion, named dfa-swb and dfa-swbLK. In the process they also generated a large deletion of 4kb of N, which removes the upstream region, first exon and part of the first intron (dN) and a rescued version bringing back this entire region (N-resc), in toto 8 new Notch alleles. Their new N alleles came in two versions, one (called [w+]) containing a mini-w+ artificial gene in the first intron of N and another, where this mini-w+ had been floxed out (called [w-]). The authors report the phenotypes of the new alleles in homozygous state as well as heterozygous over the starting deletion dN. An interesting observation is that the [w+] versions are less functional than the [w-], suggesting that embedding a gene in the first N intron compromises N expression (probably at a transcriptional or splicing step). In brief, the d1-d4 [w-] deletions behave like the wild-type and only one of the fa-swb mimics (dfa-swb[w-]) shows a hypomorphic phenotype (pupal lethality). In contrast, the [w+] versions show various degrees of eye malformations, wing nicks, bristle mispatterning and leg shape defects – some of them even show significantly reduced viability. The more severe of the small deletions, d3, overlaps a region where binding of architectural factors CTCF and Chro has been described. Indeed d3[w-] and larger deletions remove CTCF and Chro immunoreactivity from the N upstream region of polytene chromosomes and in fact remove the 3C6-3C7 interband region.
This represents a large amount of work and gives interesting new insights on how N, a presumably constitutively expressed gene, is transcriptionally regulated, implicating a chromatin organizing center as playing some role. Its role can however be only detected in a sensitized background, represented by the [w+] series of alleles. The insulator deletion in a [w-] background behaves wt. In addition to the insulator, the upstream (fa-swb) region is also important, as well as a putative CCAAT box near the TSS. Given the fact that many deletions showed no phenotype, but some did under certain conditions, the results have to be reorganized in a better way. As it stands now, it was difficult to follow the text of the paper and to reach the above conclusions owing in large part to the disorganized presentation and lack of data in tabulated form. My recommendations are the following:
First, show data that dN[w-] and [w+] are equally severe and null (compare with a larger N deletion). In presenting these data, tell the reader how many aminoacids besides the signal peptide are deleted in dN.
Describe the dominant and recessive (in hemizygotes and heterozygotes over a null allele, where possible) phenotypes of each of the deletions with or without the [w+]. Describe viability, wing notching and eye roughness for all combinations (and bristles and legs for selected combinations). Organize the text per category of phenotype for all alleles, eg dominant phenotypes of all eight new alleles. Phenotypes of hemizygous males for all eight new alleles, and so on. This will enable the reader to compare their phenotypic severity. Use tables to summarize all phenotypes.
Point out clearly in the text the data that prove that d1-d3[w+] are more severely affected than N-resc[w+] (and thus implicate the insulator in regulating N).
In Fig 7 only fa-swb (which of the three versions?), N-resc[w+] and d3[w+] are shown. It would be good to also show d1[w+] and d2[w+], which are claimed in the text NOT to lose the Chro and CTCF binding. They should also show [w-] versions of N-resc vs d3 (at least) to make sure that the band of Chro and CTCF immunoreactivity on the polytene chromosomes is not an artifact of the [w+] minigene (it is conceivable that the minigene sequences may recruit CTCF and Chro).
In the discussion, speculate how an insulator plus an upstream (fa-swb) element could be transcriptionally controlling N. In its present form the discussion is unclear in places (eg lines 498-506 and lines 517-520).
Fix the language; it has mistakes in several places (e.g. the last sentence of the abstract)
Some minor points that should be addressed:
line 239 "… various fragments of the 5'UTR of the Notch gene…" The deletions were in the upstream non-transcribed region, not in the 5' UTR.
line 309 "triplo array" I think the correct term is "triple row"
Figure 3C: there are two wings shown per genotype: are these simply two random examples or do they represent the range of the phenotypes obtained? (state it in the legend)
line 343: "…the phenotypes are normal". Does normal mean "notched wings"? (the sentence describes phenotypes of heterozygous alleles over a presumed null)
line 403: Why does the mini-w+ transgene generate a new puff upon heatshock? Does the hsp70 promoter contain the HSEs? (heatshock elements). Normally only the basal hsp70 promoter is found in most plasmids. Please comment.
line 522: spl is not an allele caused by defects in the second intron; it is a missense mutation in the EGF14 region of the coding sequence.
Author Response
Thank you very much for your constructive and useful comments. We tried our best to take your recommendations into account. This greatly improved the presentation of our data.
Q: First, show data that dN[w-] and [w+] are equally severe and null (compare with a larger N deletion). In presenting these data, tell the reader how many aminoacids besides the signal peptide are deleted in dN.
A: In the dN [w-] and [w +] deletions, the regulatory (promoter) region of the gene, all known transcription initiation sites, the first exon, including the 5'UTR and the signal peptide sequence and the subsequent 9 amino acid residues, as well as about 1 kb of the first intron, were completely deleted. The Notch gene has no alternative promoters and, therefore, we have good reason to believe that the gene product is not produced, that is, dN is a null allele (added to the text). line 229-236
Q: Describe the dominant and recessive (in hemizygotes and heterozygotes over a null allele, where possible) phenotypes of each of the deletions with or without the [w+]. Describe viability, wing notching and eye roughness for all combinations (and bristles and legs for selected combinations). Organize the text per category of phenotype for all alleles, eg dominant phenotypes of all eight new alleles. Phenotypes of hemizygous males for all eight new alleles, and so on. This will enable the reader to compare their phenotypic severity. Use tables to summarize all phenotypes.
A: Phenotypes of males hemizygous for the deletions obtained are shown in Fig.2 and are described in the text. Description of viability, wing notching, and eye roughness for some of the heterozygous combinations is presented in the Table 1. A table summarizing the phenotypic data for the alleles (deletions) obtained is now available as a Supplementary Table S1.
Q: Point out clearly in the text the data that prove that d1-d3[w+] are more severely affected than N-resc[w+] (and thus implicate the insulator in regulating N).
A: Since we observe a stronger mutant phenotype (rough eyes) in the d1-d3 [w +] lines compared to N-resc [w +] (Fig.2, Tabl S1), we believe that they are indeed more severely affected (added to the text). line 291-293
Q: In Fig 7 only fa-swb (which of the three versions?), N-resc[w+] and d3[w+] are shown. It would be good to also show d1[w+] and d2[w+], which are claimed in the text NOT to lose the Chro and CTCF binding. They should also show [w-] versions of N-resc vs d3 (at least) to make sure that the band of Chro and CTCF immunoreactivity on the polytene chromosomes is not an artifact of the [w+] minigene (it is conceivable that the minigene sequences may recruit CTCF and Chro).
A: The Fig.7 has been changed, it is added options without the AE element (resc [w-] and d3 [w-]). The fact that there is no protein immunolocalization signal in the d3 [w +] line in this region indicates that mini-white does not bind either CTCF or Chro and so it does not result in an artifactual signal. Immunostaining data on d1 [w +] and d2 [w +] are completely identical to the N-resc [w +], so we decided not to provide these images so as not to overload the figure.
Q: In the discussion, speculate how an insulator plus an upstream (fa-swb) element could be transcriptionally controlling N. In its present form the discussion is unclear in places (eg lines 498-506 and lines 517-520).
It has previously been suggested that there is an interaction between enhancers located in the introns of the Notch gene and regulatory elements of the promoter zone (Ramos et al., 1989; Vazquez and Schedl, 2000). Recently, using the Hi-C method (Stadler et al, 2017), a physical interaction was shown between the promoter region of the Notch gene and the sequence in the body of this gene with TAD formation of about 20 kb. It can be assumed that if AE is inserted into the first intron, physical interactions between the hypothetical enhancer and the insulator sequence in the promoter region of the gene are disrupted. Further experiments using the Hi-C method can help test this hypothesis using the deletions we obtained. Appropriate changes have been introduced into the text of discussion. line 522-529
Q: Fix the language; it has mistakes in several places (e.g. the last sentence of the abstract)
A: We checked the text more carefully and corrected all the errors detected.
Some minor points that should be addressed:
Q: line 239 "… various fragments of the 5'UTR of the Notch gene…" The deletions were in the upstream non-transcribed region, not in the 5' UTR.
A: Changes were made to the text
Q: line 309 "triplo array" I think the correct term is "triple row"
A: Done
Q: Figure 3C: there are two wings shown per genotype: are these simply two random examples or do they represent the range of the phenotypes obtained? (state it in the legend)
A: The range of the phenotypes obtained is shown (changes have been made to the text)
Q: line 343: "…the phenotypes are normal". Does normal mean "notched wings"? (the sentence describes phenotypes of heterozygous alleles over a presumed null)
A: By normal, we mean wild-type phenotype (corrected) line 403:
Q: Why does the mini-w+ transgene generate a new puff upon heatshock? Does the hsp70 promoter contain the HSEs? (heatshock elements). Normally only the basal hsp70 promoter is found in most plasmids. Please comment.
A: The miniwhite gene is controlled by the hsp-70 promoter. The pGE-attB-GMR construct (Huang et.al., 2009) contains a full-sized hsp70 promoter (559 bp) that contains all the necessary HSEs (Simon and Lis, 1987). Mini-white lack its own promoter, and its coding sequense is linked downstream of the hsp-70 promoter. It has been reported in multiple studies that such hybrid constructs (in which hsp-70 promoter drives expression of the gene of interest) form a puff in polytene chromosomes (for example, please see Semeshin et al, 1989).
Q: line 522: spl is not an allele caused by defects in the second intron; it is a missense mutation in the EGF14 region of the coding sequence.
Мutation facet(fa) affects eye morphology and is associated with the insertions of transposable elements in the second intron of Notch[18] (changes have been made to the text)